# Telehealth and Burn Care: From Faxes to Augmented Reality

**DOI:** 10.3390/bioengineering9050211

**Published:** 2022-05-13

**Authors:** Caroline Park, Youngwoo Cho, Jalen Harvey, Brett Arnoldo, Benjamin Levi

**Affiliations:** Department of Surgery, University of Texas Southwestern Medical Center, Dallas, TX 75390, USA; youngwoo.cho@utsouthwestern.edu (Y.C.); jalen.harvey@utsouthwestern.edu (J.H.); brett.arnoldo@utsouthwestern.edu (B.A.); benjamin.levi@utsouthwestern.edu (B.L.)

**Keywords:** telemedicine, telehealth, augmented reality, burn wound, burn surgery

## Abstract

Despite advances in telemedicine, practices remain diverse, ranging from telephonic to still images and video-based conferencing. We review the various modes of telemedicine in burn care and summarize relevant studies, including their contributions and limitations. We also review the role of a more recent technology, augmented reality, and its role in the triage and management of burn patients. Telemedicine in burn care remains diverse, with varied outcomes in accuracy and efficiency. Newer technologies such as augmented reality have not been extensively studied or implemented but show promise in immersive, real-time triage.

## 1. Introduction

Despite technological advances in telemedicine, initial burn triage and care remain diverse, ranging from still images on phone-based applications or electronic medical records (“EMR”) to more advanced, real-time video-based technologies. Reports indicate that clinicians fail to make the correct diagnosis 40% of the time when assessing burn wound depth [1,2,3]. This inability to distinguish wound depth and extent can cause either inadequate tissue excision, leading to wound infection and wound failure, or excessive debridement and inappropriate tissue removal, leading to iatrogenic injury. However, telemedicine is no stranger to burn surgery, with many centers throughout the United States utilizing anything from “store and forward” to interactive video communication [4,5]. There are several contributing factors to this variety in communication, including: (1) non-standardized image-capture technology and EMR, (2) varied costs of and rate of change in technology and (3) buy-in from providers to learn and adopt. Many burn wounds are low in total body surface area (“TBSA”) and in severity but are still over-triaged and transferred to burn centers with associated costs to the patient and the healthcare system [6,7]. We review the various modalities used for burn-wound triage and treatment and propose an augmented reality (“AR”), hands-free, head-mounted display (ARHMD) to help safely triage burn wounds in real time with the goal to decrease over-triage and improve accuracy in burn TBSA and severity.

## 2. Background

### 2.1. Cameras and Facsimiles

Burn wound triage in TBSA and burn depth requires training with repetition and experience, which occurs at higher-level burn centers. Burn wounds, in addition to cutaneous malignancies and skin infections, are highly amenable to image-based technology because of their visible nature, ease of reproducing images, and need to monitor progress as part of treatment. One of the earlier, published series in the military consisted of photos and facsimiles as part of an overseas tele-health initiative [8]. These case reports included one of a chemical burn to the face and cornea, shrapnel to the chest, and other ophthalmologic cases. These early communications mark the beginning of advanced tele-health within the military. However, there were several limitations, including quality of images, decreased or varied abilities to assess depth, severity, and TBSA with still images, and frequency of communication. The assessment of burn injuries is a dynamic process that requires serial assessment to evaluate for progression of disease to mitigate complications as infection and scarring.

### 2.2. Hand-Held Phone and Application-Based Technology

Once hand-held phones became prevalent, phone-based documentation replaced bulky cameras and facsimiles, but relied on text messaging (SMS), application-based messaging and e-mail to upload images prior to discussing the case with the receiving burn surgeon. Several of these studies found improvement in safely down-triaging patients [7,9,10,11]. Photo-based triage was also found to have the same inter-reliability amongst surgeons performing face-to-face examinations compared to image-based via application-based messaging [12]. However, these modalities may not be the most secure and produce concerns about medico-legal practices and security of patient information [13]. So-called “store and forward” telemedicine could address this security concern by uploading an image to an encrypted cloud-based service or by secure e-mail, which then would be received by the burn or plastic and reconstructive surgeon [14]. However, store and forward typically does not occur in real time during patient triage and early resuscitation, leading to some delay.

### 2.3. Shared Electronic Medical Record and Cloud-Based Services

The shared electronic medical record with ‘in-network’ or collaborating facilities has facilitated the sharing of essential patient information without compromising patient security. However, several limitations remain, including a lack of real-time viewing and interaction, and the costs to networks to build these virtual bridges and maintain these data [15,16].

### 2.4. Video-Based Burn Tele-Medicine

One of the more overlooked details in initial burn management are dynamic, including the patient’s appearance, pain, vital signs and response to treatment. Images of burn wounds may not necessarily capture blanching or be sufficiently detailed to look for the presence of hair buds and other signs indicating depth of burn. Video-based telemedicine has been utilized to address some of these concerns. In one study, video was used to record the burn wound, including TBSA, and initial management [17]. This modality did improve burn size estimation and management; however, the video was “stored and forwarded,” thus not analyzed in real-time.

Real-time, interactive video has been used for burn patients, but mostly for after-care with good success with no re-admissions or complications and cost savings, including travel distance and travel time with overall cost reductions to the patient [18]. This mode does require a computer or laptop with a high-definition camera for both parties, and call-routing software and has been found to be more time-consuming for providers.

### 2.5. Augmented Reality and Hands-Free Telemedicine

There are a few modes of “realities”, ranging from completely immersive and lack of a “see-through” interface, as demonstrated in Figure 1 and Figure 2, to an augmented reality in Figure 3, where the interface is see-through but with super-imposed images. The most concise definition is a ‘reality-virtuality continuum’ between the real and virtual worlds, where virtual is less ‘real’ and augmented reality approaches the real world [19]. The most commonly known and popular interactive “smart” hands-free devices were smart glasses, which were used as part of a wound evaluation and management program. These glasses incorporated a camera with a real-time, interactive platform for performing wound dressings and operations in patients with lower extremity wounds [20].

To our knowledge, augmented reality on a hands-free, head-mounted display has not been utilized and studied on burn patients for tele-mentoring and tele-medicine. The Hololens (Microsoft©, Redmond, WA, USA) is a ‘hands-free’ display with processing speeds equivalent to a computer with a processor of 1.1 GHz and minimum hard drive space of 3.0 GB and RAM of 4.0 GB. Real-time conferencing requires Wi-Fi speeds of around 100 Mbps, with an average 1080 p video requiring 1.5 Mbps. These elements are essential to real-time and high-quality triage.

## 3. Discussion

At many teaching hospitals, the junior resident may observe several burn wounds under faculty supervision before they achieve competency. This learning paradigm has been negatively affected by a reduction in trainee work hours, which are now limited to 80 hours despite a subjective decrease in clinical performance [21]. Despite efforts to improve surgical training [22], human error from trainees and non-experienced surgeons can pose risks of burn over- or under-triage, including over- or under-resuscitation and their subsequent complications including infection, secondary organ failure and hypertrophic scarring. Thus, supervision must be high-yield and high-quality.

In response, the teaching paradigm has evolved to include not only basic, in-person supervision, but also virtual simulation and remote-assist with augmented technology. Virtual simulation, however, while immersive, is not interactive and does not allow the trainee to transfer skills and communicate in real time with others [23]. Augmented reality combines the immersive aspect of virtual reality with real-time application, including the onlay of images [24] and viewing of multiple platforms—anatomic modules [25], live-streaming video, and other resources.

## 4. Conclusions

Telementoring and tele-health have come a long way in technology, quality, time to diagnosis and delivery of care, with a decrease in patient transfers, bed days and costs associated with unnecessary transfers [26]. Several reviews on this subject exist that summarize the utilization and outcomes of telehealth in burns [4,27] and telehealth in surgery [28] without clear consensus on standardized modalities or a standard of care.

The implementation of augmented reality in burn care is the future in this era of virtual conferencing and communication, and to our knowledge has not been implemented or studied on a large scale. Two major areas should be addressed to successfully translate augmented reality technology to telementoring and tele-triage, including: (1) ensure real-time, remote-assist feedback during procedures, and (2) improve accuracy and diagnosis with the ultimate goal of improving patient care, increasing efficiency and decreasing costs to the patient and system.

## Figures and Tables

**Figure 1 bioengineering-09-00211-f001:**
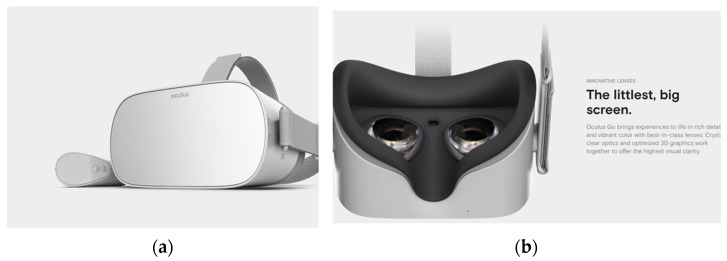
Sample individual virtual-reality headset (**a**) front and (**b**) user interface.

**Figure 2 bioengineering-09-00211-f002:**
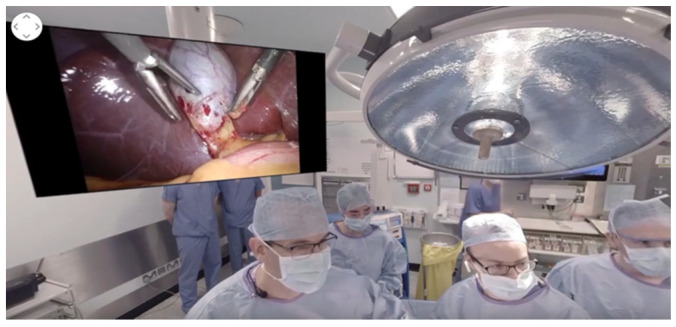
Sample screen (intra-operative view of laparoscopic cholecystectomy). Laparoscopic Cholecystectomy 360 Video—Professor Arnie Hill (Surgical Affairs, RCSI, March 2017). https://www.youtube.com/watch?v = 1vw1Z21v1EU (accessed on 22 January 2022).

**Figure 3 bioengineering-09-00211-f003:**
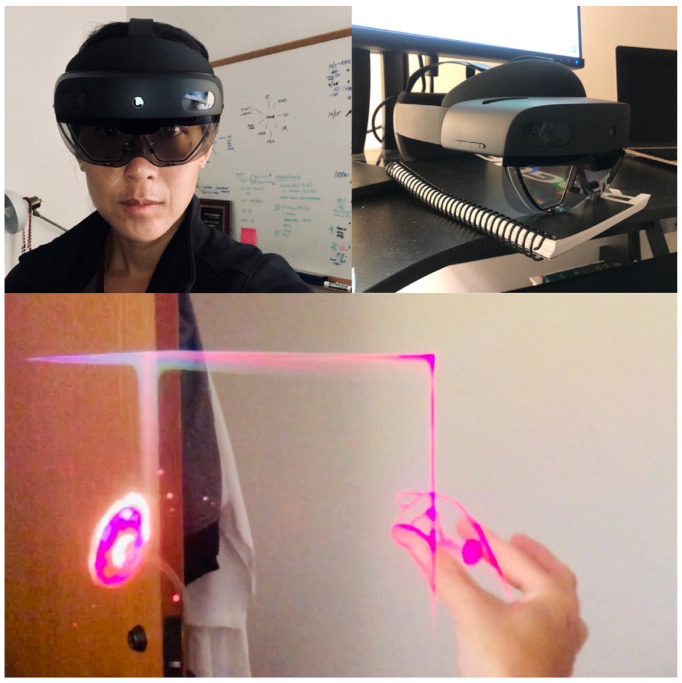
Sample augmented reality headset (hands-free, head-mounted display). User is wearing the device and projected image is displayed in real time in a real environment.

## Data Availability

Not applicable.

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
