# Peer review of "Telehealth and Burn Care: From Faxes to Augmented Reality"

_bioengineering, 2022, doi:10.3390/bioengineering9050211_

Round 1

Reviewer 1 Report

Dear Authors,

I would like to congratulate you on the idea for conducting the aforementioned study. The implementation of technologies, which allow granting health benefits at a distance is a challenge today, similarly to increasing the safety of a patient, using augmented reality technology. These technologies have already been developed in such a way that allows them to be used in modern medicine, especially with the lacking amount of medical staff at various levels. I would like to suggest, however, a slight addition to this article, which could enable other medical professionalists to also use this path:

  1. At least in a general outline, it would be good to indicate the necessary technical conditions e.g. the speed of the connection, the power of the processors etc.
  2. It is necessary to indicate which measured parameters will be indicative of the success of this study. Do you plan to check the usefulness of the used technology in the opinion of the direct users in any way? Do you intend on gathering comments connected to the issues, which should be improved? If the answer is affirmative then I would ask you to describe the methods of evaluation and the ratings of the project in the article.
  3. In what way will the level of the acquired knowledge (the correctness of the actions), realized by the younger, studying staff be validated, checked? Please include a paragraph concerning this topic.

Author Response

  1. At least in a general outline, it would be good to indicate the necessary technical conditions e.g. the speed of the connection, the power of the processors etc. 
    1. Thank you. We have provided these technical specifications in section 2.5 on page 3. 
  2. It is necessary to indicate which measured parameters will be indicative of the success of this study. Do you plan to check the usefulness of the used technology in the opinion of the direct users in any way? Do you intend on gathering comments connected to the issues, which should be improved? If the answer is affirmative then I would ask you to describe the methods of evaluation and the ratings of the project in the article.
    1. We have incorporated feedback from other reviewers that the paper cannot published without preliminary data, thus we have removed our proposed study. 
  3. In what way will the level of the acquired knowledge (the correctness of the actions), realized by the younger, studying staff be validated, checked? Please include a paragraph concerning this topic.
    1. Thank you for this question. We have decided to remove the proposed study given other reviewers' comments. To answer your question, we plan to use objective structured assessment tools (OSATs) to evaluate proficiency and execution.

Reviewer 2 Report

The authors propose an Augmented Reality (AR) system in telehealth and burn care, a topic of interest for the journal. However, while medical applications based on AR are promising, the work presented in the current paper is just a proposal, not a real implementation and validation of the results. Furthermore, in such a scenario, the conclusions can be considered only hypothesis, authors need to prove their claims. For these reasons, I cannot recommend this paper for publication.

Author Response

The authors propose an Augmented Reality (AR) system in telehealth and burn care, a topic of interest for the journal. However, while medical applications based on AR are promising, the work presented in the current paper is just a proposal, not a real implementation and validation of the results. Furthermore, in such a scenario, the conclusions can be considered only hypothesis, authors need to prove their claims. For these reasons, I cannot recommend this paper for publication.

Thank you for this evaluation. We respect your uncompensated time in reviewing our manuscript. We have removed the proposed study from the manuscript and revised it to be a narrative review on this topic.

Reviewer 3 Report

Dear Sirs,

In a present form I unfortunately had to suggest rejection of Your paper. The reason is lack of clear current results of research. I found in the paper just an idea for applying augmented reality and plan of research, in order to increase ergonomic of transferring an online video stream or a picture from HMDs instead of using smartphone camera, enable specialists consulting  for optimal triage and minimize costs of healthcare. I am aware such a study needs time and a lot of efforts, but at the beginning of this journey I believe some reasonable tests on selected HMDs models taking into account aspects of: clear, good contrast visibility of burn wounds in any conditions, their resolutions, problem of latency,  method of color/size calibration, etc. would be a valuable development of Your present paper.  An important issue would be also if the triage would be automated by a decision supporting algorithm or if Your system’s task is only to transfer the data and the complete responsibility has the tele-consultant.  It is worth answering these questions thinking on scientific aspect of Your research .

Thank You for ability to review Your work and I wish You success in Your further research.

Kind regards

Author Response

In a present form I unfortunately had to suggest rejection of Your paper. The reason is lack of clear current results of research. I found in the paper just an idea for applying augmented reality and plan of research, in order to increase ergonomic of transferring an online video stream or a picture from HMDs instead of using smartphone camera, enable specialists consulting  for optimal triage and minimize costs of healthcare. I am aware such a study needs time and a lot of efforts, but at the beginning of this journey I believe some reasonable tests on selected HMDs models taking into account aspects of: clear, good contrast visibility of burn wounds in any conditions, their resolutions, problem of latency,  method of color/size calibration, etc. would be a valuable development of Your present paper.  An important issue would be also if the triage would be automated by a decision supporting algorithm or if Your system’s task is only to transfer the data and the complete responsibility has the tele-consultant.  It is worth answering these questions thinking on scientific aspect of Your research .

Thank you for your review and time. We agree that we should include preliminary data and have removed the study proposal from this manuscript. We have instead approached the manuscript as a narrative review. 

Round 2

Reviewer 1 Report

I accept the introduced amendments, I have no more comments.

Author Response

Thank you for your comments and your time.

Reviewer 3 Report

Dear Sirs,

After changing the scope to narrative review I accept the paper. 

Kind regards

Author Response

The authors greatly appreciate your time.